# Bacterial DNA on the skin surface overrepresents the viable skin microbiome

Ellen M Acosta[1], Katherine A Little[1], Benjamin P Bratton[1,2,3,4], Jaime G Lopez[2], Xuming Mao[5], Aimee S Payne[5], Mohamed Donia[1], Danelle Devenport[1], Zemer Gitai[1]*

[1]Department of Molecular Biology, Princeton University, Princeton, United States; [2]Lewis-Sigler Institute for Integrative Genomics, Princeton University, Princeton, United States; [3]Department of Pathology, Immunology and Microbiology, Vanderbilt University Medical Center, Nashville, United States; [4]Vanderbilt Institute for Infection, Immunology and Inflammation,, Nashville, United States; [5]Department of Dermatology, University of Pennsylvania, Philadelphia, United States

**Abstract** The skin microbiome provides vital contributions to human health. However, the spatial organization and viability of its bacterial components remain unclear. Here, we apply culturing, imaging, and molecular approaches to human and mouse skin samples, and find that the skin surface is colonized by fewer viable bacteria than predicted by bacterial DNA levels. Instead, viable skin-associated bacteria are predominantly located in hair follicles and other cutaneous invaginations. Furthermore, we show that the skin microbiome has a uniquely low fraction of viable bacteria compared to other human microbiome sites, indicating that most bacterial DNA on the skin surface is not associated with viable cells Additionally, a small number of bacterial families dominate each skin site and traditional sequencing methods overestimate both the richness and diversity of the skin microbiome. Finally, we performed an in vivo skin microbiome perturbation-recovery study using human volunteers. Bacterial 16S rRNA gene sequencing revealed that, while the skin microbiome is remarkably stable even in the wake of aggressive perturbation, repopulation of the skin surface is driven by the underlying viable population. Our findings help explain the dynamics of skin microbiome perturbation as bacterial DNA on the skin surface can be transiently perturbed but is replenished by a stable underlying viable population. These results address multiple outstanding questions in skin microbiome biology with significant implications for future efforts to study and manipulate it.

## eLife assessment

In this **important** study, the authors provide **convincing** evidence that current DNA-based microbial genomics for skin bacteria cannot always detect the source of sequenced DNA and whether it originated from viable or non-viable bacteria. Additionally, the authors demonstrated in humans and mice that most of the viable bacteria reside inside hair follicles rather than the surface of the skin per se. Overall, the work has significance beyond a single discipline and will be of interest to those studying microbiomes.

## Introduction

The skin is the largest organ in the human body, providing roughly 25 square meters for potential host–microbe interactions (*Gallo, 2017*). It facilitates our tactile interactions with the world, separates

*For correspondence:
zgitai@princeton.edu

us from the dangers of our daily lives, and has the incredible ability to regenerate itself every 20–30 d (*Maeda, 2017*). Human skin is also home to the organisms that comprise the skin microbiome, which has been shown to have important roles in human health. For example, the human skin microbiome affects immune system education (*Scharschmidt et al., 2015*; *Polak-Witka et al., 2020*), wound healing (*Kalan et al., 2019*; *Kim et al., 2019*; *Loesche et al., 2017*), colonization resistance (*Byrd et al., 2018*), modulation of gene expression in the skin (*Meisel et al., 2018b*), and may have a role in development (*Meisel et al., 2018a*). Despite the many contributions of the skin microbiome to human health, there are important questions that have not been addressed by traditional methods: Why is the skin microbiome stable across months in longitudinal studies yet easily perturbed upon transient environmental changes like swimming (*Nielsen and Jiang, 2019*)? Why are there so many anaerobic bacteria on an organ exposed to the air? And why is it so difficult to stably colonize the skin with new microbes without strong perturbations like abrasion? Traditional methods of sampling, culturing, and sequencing the bacterial members of the skin microbiome have been indispensable for establishing the skin microbiome field. Here, we extend these foundational studies with additional approaches to differentiate between viable and nonviable bacteria in order to begin to address these important questions.

## Results

### Fluorescence in situ hybridization reveals few bacteria on the skin surface

The predominant method by which skin microbiomes have been studied is through sequencing of DNA from swabbed skin areas (*Huttenhower et al., 2012*). However, despite the fact that the skin microbiome is often depicted as a well-mixed coating of microbes on the skin surface (*Byrd et al., 2018*; *Grice and Segre, 2011*), it remains unclear whether the DNA from the skin surface reflects the underlying biology of the skin microbiome. Additionally, while sequencing and culture-based studies have demonstrated that bacteria extend into deeper portions of the skin, the spatial distribution of the skin microbiome as a whole has not been well characterized (*Polak-Witka et al., 2020*; *Belkaid and Tamoutounour, 2016*; *Lousada et al., 2021*). To address this gap, we determined the spatial distribution of bacterial cells in the skin using the universal bacterial fluorescence in situ hybridization (FISH) probe EUB338, which hybridizes to bacterial 16S rRNA (*Amann et al., 1990*).

We first used EUB338 FISH on biopsied healthy adult human facial tissues and found that the skin surface contains very few bacteria (*Figure 1A*). In contrast, clusters of bacteria were found within hair follicles and other cutaneous skin structures like comedos (*Figure 1B*). There are several possible explanations for the lack of FISH staining on the skin surface. For example, viable surface bacteria may have been eliminated by sterilization of the biopsied area (we address this possibility below). Alternatively, the surface bacteria could exist in a state like stationary phase that has fewer EUB388-hybridizable ribosomes, or by skin microbiome bacterial species being less prone to FISH probe hybridization. However, we confirmed that EUB338 staining works well in stationary-phase cells of multiple skin microbiome bacterial species (*Cutibacterium acnes, Staphylococcus epidermidis, Micrococcus luteus,* and *Corynebacterium striatum*) (*Figure 1—figure supplement 1*).

To quantify our FISH staining of human skin, we calculated the ratio of the mean fluorescence within an area of interest (a hair follicle, the skin surface, or other cutaneous structures) to the mean fluorescence outside of the area of interest. We refer to this ratio as an enrichment score. The median enrichment score for human follicles was 11.24 (*Figure 1D*). In contrast, the skin surface (stratum corneum) had an enrichment score of just 0.188. This pattern of observing many bacteria in follicles but few on the skin surface held true when using FISH probes specific for *C. acnes*, one of the most abundant bacterial species in the human skin microbiome (*Figure 1C*). For *C. acnes*, the enrichment score was 3.14 for follicles compared to 0.67 for the stratum corneum (*Figure 1D*). We note that while the majority of intact skin-associated bacteria were not associated with the skin surface, there are some visible bacteria on the skin surface, especially in areas near subsurface structures (*Figure 1A and B*). This indicates that FISH staining is capable of detecting bacteria on the skin surface. Thus, our results are consistent with previous reports that bacteria can be cultured from skin surface swabs but also extend these findings to demonstrate that the skin surface has fewer intact bacteria than deeper skin structures.

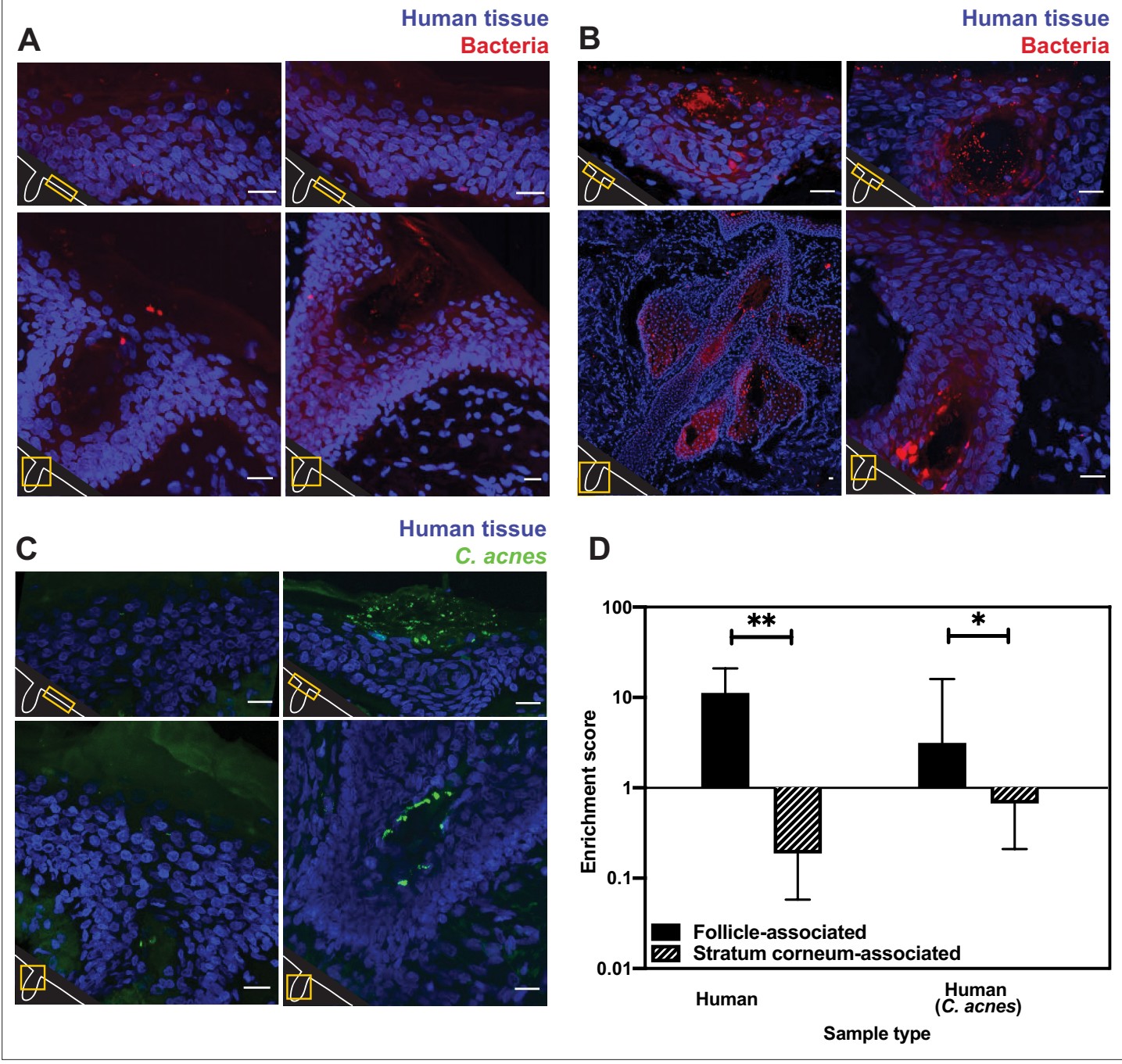

**Figure 1.** Bacterial fluorescence in situ hybridization (FISH) staining of human tissue. (**A–C**) Scale bar = 20 µm. The bottom-left corner of each diagram shows a schematic of the hair follicle in white and the anatomic location of each image frame in yellow. DAPI staining is shown in blue in all parts. EUB338 hybridization is shown in red for all images. (**A**) Human tissues stained with the pan-bacterial FISH probe EUB338 show little bacterial presence at the skin surface. (**B**) Human tissues stained with EUB338 show abundant bacterial signal that is concentrated in hair follicles, pilosebaceous units, and other cutaneous structures. (**C**) Human tissues stained with a *C. acnes*-specific FISH probe (in green) demonstrate the same overall spatial organization as those stained with EUB338. (**D**) Quantification enrichment scores showing the median and interquartile range. Significance was calculated using the Mann–Whitney test. *p≤0.05, **p≤0.01. N = 8 for human follicle, N = 6 for human follicle (*C. acnes*), N = 6 for human stratum corneum, N = 5 for human stratum corneum (*C. acnes*) where 'N' represents different follicles or stratum corneum sections. The human tissue samples shown here were obtained from adult facial tissues (cheek and forehead).

The online version of this article includes the following figure supplement(s) for figure 1:

**Figure supplement 1.** Fluorescence in situ hybridization (FISH) on stationary-phase skin microbiome bacterial species.

## Bacterial DNA on the skin surface overrepresents the number of viable bacteria

DNA sequencing of samples collected by swabbing the skin surface is the most common method used to evaluate the skin microbiome because it is simple, noninvasive, and has been shown to result in higher consistency than other sampling methods (*Bjerre et al., 2019*). However, traditional metagenomic and 16S rRNA gene sequencing do not differentiate DNA from viable and inviable bacteria. We thus implemented a method that allowed us to quantitatively determine the fraction of DNA in a sample that is specifically from intact bacteria. The ability to molecularly differentiate between viable and inviable cells would enable us to both validate our FISH results using an orthogonal method on untreated, living skin, and address the possibility that the low numbers of intact surface bacteria observed by FISH were due to biopsy sterilization. To this end, we utilized the cell-impermeable small molecule propidium monoazide (PMA), which binds irreversibly to double-stranded DNA upon photoactivation to inhibit PCR amplification (*Figure 2A*; *Nocker et al., 2006*). When PMA photoactivation is performed before the cell lysis step of DNA isolation, the genomic DNA inside viable bacteria is protected from PMA binding because PMA is cell-impermeable, while cell-free DNA or DNA within permeabilized bacteria becomes PMA-bound. To quantify bacterial DNA, we combined the use of PMA with droplet digital PCR (PMA-ddPCR). Calculating the ratio of ddPCR counts between samples without PMA and samples with PMA allowed us to generate a viability score for any given bacterial population. A similar approach has been used to assess the viability of bacteria from environmental waste water samples (*Yang et al., 2017*).

To assure that PMA-ddPCR would allow us to reliably gauge the fraction of viable cells in a population, we first validated that it generated the expected results using known ratios of either exponentially-growing or stationary-phase *Escherichia coli* cultures combined with heat-killed *E. coli* cells (*Figure 2—figure supplement 1A*). To determine whether PMA-ddPCR also works for skin-resident bacterial species, we also applied PMA-ddPCR to four of the most common skin microbiome bacteria, *S. epidermidis, C. acnes, M. luteus,* and *C. striatum*. We performed serial dilutions of each bacterial culture and in parallel performed both colony-forming units (CFU) plating and ddPCR with or without PMA. In every case, the amount of DNA in a sample (assessed by PMA-ddPCR) correlated well with the number of culturable bacteria in high and low bacterial abundance scenarios (*Figure 2—figure supplement 1B*). Together, these controls confirm that PMA-ddPCR is a reliable method for assessing the amount of DNA in a sample present within intact bacteria.

We next applied PMA-ddPCR to human skin microbiome samples by swabbing the skin of four healthy human volunteers at eight sites (glabella, retroauricular crease, lower back, hair shaft, antecubital fossa, popliteal fossa, nares, and dorsal forearm) (*Figure 2B*, *Figure 2—figure supplement 2A and B*). PMA-ddPCR revealed that the viability scores for these sites ranged between 0.02 and 0.12 (0 represents a fully nonviable population, 1.0 represents a fully viable population), indicating that the majority of bacterial DNA found on the skin surface are not associated with viable cells (*Figure 2C*). To investigate whether this was a skin-specific phenomenon, we tested several non-skin microbiome sites (tongue, saliva, plaque, and feces). We found that in all non-skin microbiome sites, the viability score was significantly higher than for the skin, ranging from 0.4 (saliva) to 0.87 (feces) (*Figure 2C*).

We next sought to address whether our PMA-ddPCR viability scores accurately represent of the number of viable bacteria on the skin surface. The low number of detectable bacteria on the skin surface (either by FISH or PMA-ddPCR) does not imply that the skin surface is sterile but rather that the majority of the bacterial DNA on the skin surface are not from these viable cells. Indeed, culturing bacteria directly from the skin is common (*Byrd et al., 2018*). To quantify the number of viable bacteria directly, we plated a small amount of each sample using the standard conditions for culturing skin microbes (5% sheep blood in tryptic soy agarose plates incubated both aerobically and anaerobically). To determine whether PMA-ddPCR or traditional ddPCR better represented the number of viable skin microbiome bacteria, we compared our results to a standard curve generated with known numbers of *S. epidermidis*. For each sample, the PMA-ddPCR quantification closely matched this standard curve while the samples lacking PMA showed no overlap (*Figure 2—figure supplement 1C and D*). The highest abundance skin microbiome species are readily culturable, such that quantifying the DNA from viable skin surface bacteria should be able to accurately predict CFUs upon plating. Quantifying the bacterial DNA in skin microbiome samples without the use of PMA resulted in DNA quantities that were, on average, 82 times higher than predicted by the standard curve, while the use of PMA

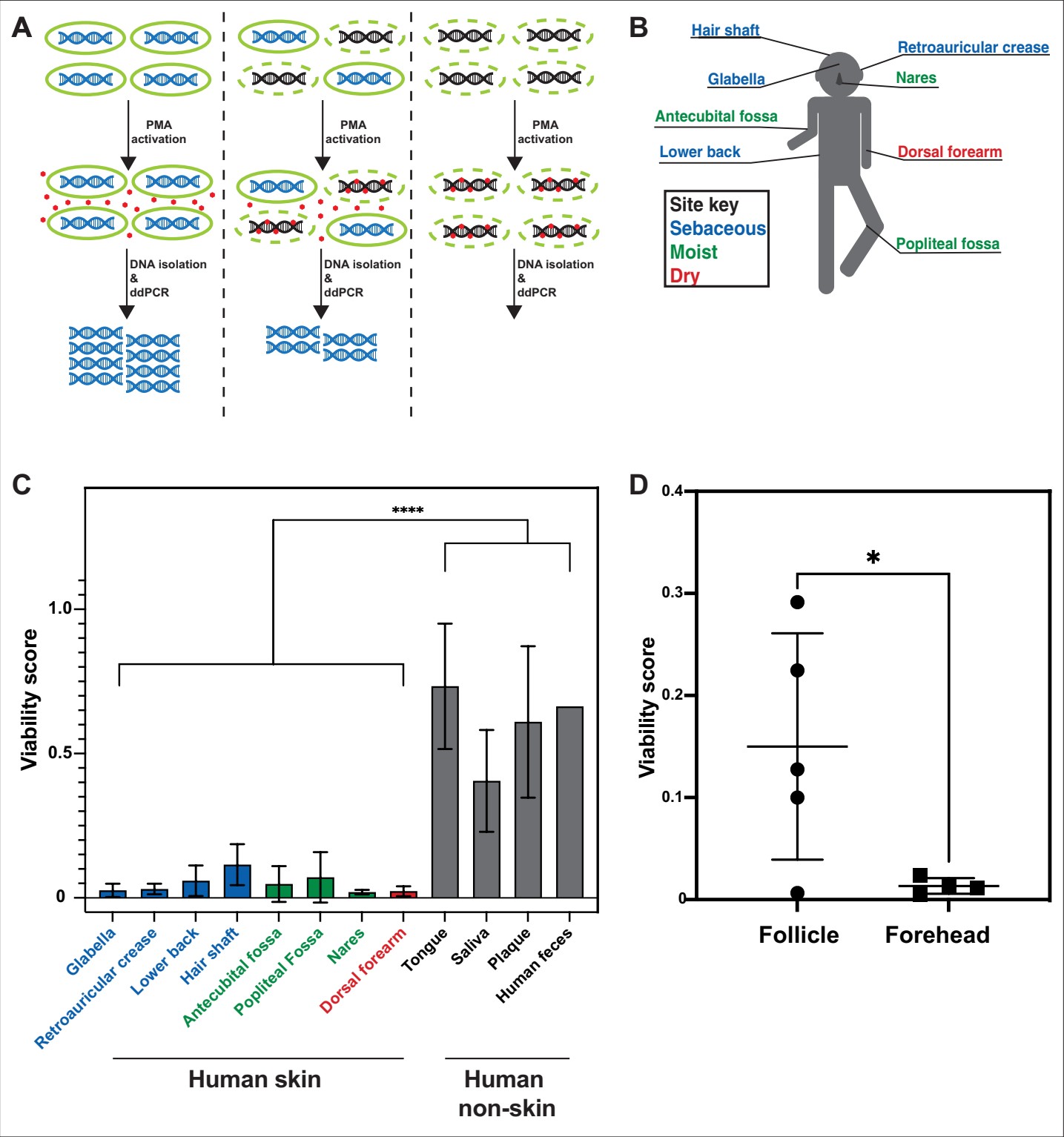

**Figure 2.** Propidium monoazide-droplet digital PCR (PMA-ddPCR) and viability scores for human skin and non-skin microbiomes. (**A**) Schematic of the PMA-ddPCR workflow. (**B**) Sampling scheme showing each skin site that was sampled. Colors indicate site type (sebaceous in blue, moist in green, dry in red). (**C**) PMA-ddPCR on skin and non-skin microbiome sites shows that the viability score of the skin microbiome is significantly lower than other microbiome sites. ****$p \leq 0.0001$ for Student's *t*-test on pooled skin and non-skin samples. Four volunteers contributed skin and non-skin microbiome samples. Additional samples were collected from some individuals and represent biological replicates. N = 8 for glabella, N = 6 for retroarticular crease, N = 5 for lower back, hair shaft, nares, and dorsal forearm, N = 3 for antecubital fossa, tongue, saliva, and plaque, N = 2 for popliteal fossa, and N =

*Figure 2 continued on next page*

*Figure 2 continued*

1 for human feces. Each human skin sample site consists of samples from four different individuals. Some volunteers were sampled multiple times on different days (at least 2 wk apart). For glabella, one volunteer was sampled four times, one volunteer was sampled two times, and two volunteers were sampled one time. For retroauricular crease, two volunteers were sampled two times, and two volunteers were sampled one time. For lower back, one volunteer was sampled two times and three volunteers were sampled one time. For hair shaft, all samples came from one volunteer. For antecubital fossa, three volunteers were sampled one time. For popliteal fossa, two volunteers were sampled one time. For nares, one volunteer was sampled two times and three volunteers were sampled one time. For dorsal forearm, one volunteer was sampled two times and three volunteers were sampled one time. Tongue, saliva, and plaque all represent one sample from three different individuals. For raw ddPCR counts, see *Figure 2—figure supplement 2A and B*. (**D**) PMA-ddPCR on follicle contents and forehead swabs from five individuals. Mean viability score for follicle contents is 0.15 and for forehead is 0.013. All error bars indicate standard deviation.

The online version of this article includes the following figure supplement(s) for figure 2:

**Figure supplement 1.** Propidium monoazide-droplet digital PCR (PMA-ddPCR) and sampling controls.

**Figure supplement 2.** Copies per 20 µL droplet digital PCR (ddPCR) reaction without (**A**) and with (**B**) the use of propidium monoazide (PMA).

brought this value down to just 1.3 (*Figure 2—figure supplement 1E*). Using ddPCR counts to predict CFU showed similar results, as ddPCR in the absence of PMA yielded values that predicted CFU counts 58.5 times greater than those measured, while PMA-ddPCR yielded values that predicted CFU counts that were on average only 1.28 times greater than the actual cultured CFU (*Figure 2—figure supplement 1F*). Since we expect a ratio near 1, these findings lend further support for our conclusion that PMA-ddPCR reflects the viable microbiome better than the approaches lacking PMA.

Our FISH results from biopsied skin suggested that viable bacteria may be protected below the skin surface. To test this hypothesis in untreated skin, we collected facial follicle contents and used PMA-ddPCR to compare the viability of facial follicle contents and skin surface samples (*Figure 2D*). The average viability for follicle contents was greater than tenfold higher for follicle contents than for the skin surface, confirming that cells in hair follicles are more viable than those on the skin surface. Together, our results from FISH on biopsies and PMA-ddPCR on skin swabs independently support the conclusion that the skin surface is populated by few viable bacterial cells, indicating that the surface of healthy, non-sterilized human skin is sparsely colonized. While these data indicate that bacterial DNA on the skin surface is predominantly not associated with viable bacterial cells, we note that our results do not suggest that there are no viable cells on the skin surface. Rather, our data indicate that the majority of the bacterial DNA on the skin surface are not within bacteria such that using PMA provides a much more accurate estimation of the viable skin microbiome.

## Traditional sequencing methods overestimate skin microbiome richness and diversity

Like many microbiomes, the existing knowledge of the skin microbiome is heavily based upon bacterial 16S rRNA gene amplicon sequencing, which was developed to assess bacterial populations while avoiding biases introduced by culturing methods. However, our findings suggest that using 16S rRNA gene amplicon sequencing to study the skin microbiome is not entirely unbiased as most of the DNA in these samples are not from viable bacteria and traditional 16S rRNA gene amplicon sequencing does not differentiate between DNA originating from live or dead cells. The inability of 16S rRNA gene amplicon sequencing to differentiate between these two types of bacterial populations has been mentioned as a potential downfall of the method (*Byrd et al., 2018*). To evaluate how accurately traditional 16S rRNA gene amplicon sequencing captures the living skin microbiome composition, we utilized PMA followed by 16S rRNA gene amplicon sequencing (an approach we refer to as PMA-seq) (*Nocker et al., 2010*). By sequencing pairs of matched samples with PMA treatment (PMA-seq) and without PMA treatment (traditional sequencing), we were able to explore how closely the microbiome compositions obtained from traditional sequencing methods resembled the viable microbiome composition obtained by PMA-seq (*Figure 3A*). These experiments established that at each skin site sampled, compared to traditional sequencing, the PMA-treated samples were less rich (richness, $R$, is a measure of the number of identifiable bacterial taxa) and less diverse (diversity, $H$, is measured by the Shannon diversity index) (*Figure 3B and C*). Furthermore, samples that had greater richness in traditional sequencing ($R_{trad}$) showed proportionally larger decreases in richness and Shannon diversity with PMA-seq ($R_{PMA}$ and $H_{PMA}$) (*Figure 3B and C*). These results suggest that, although it appears by traditional sequencing that there is a wide range of richness values at different skin sites (1–30

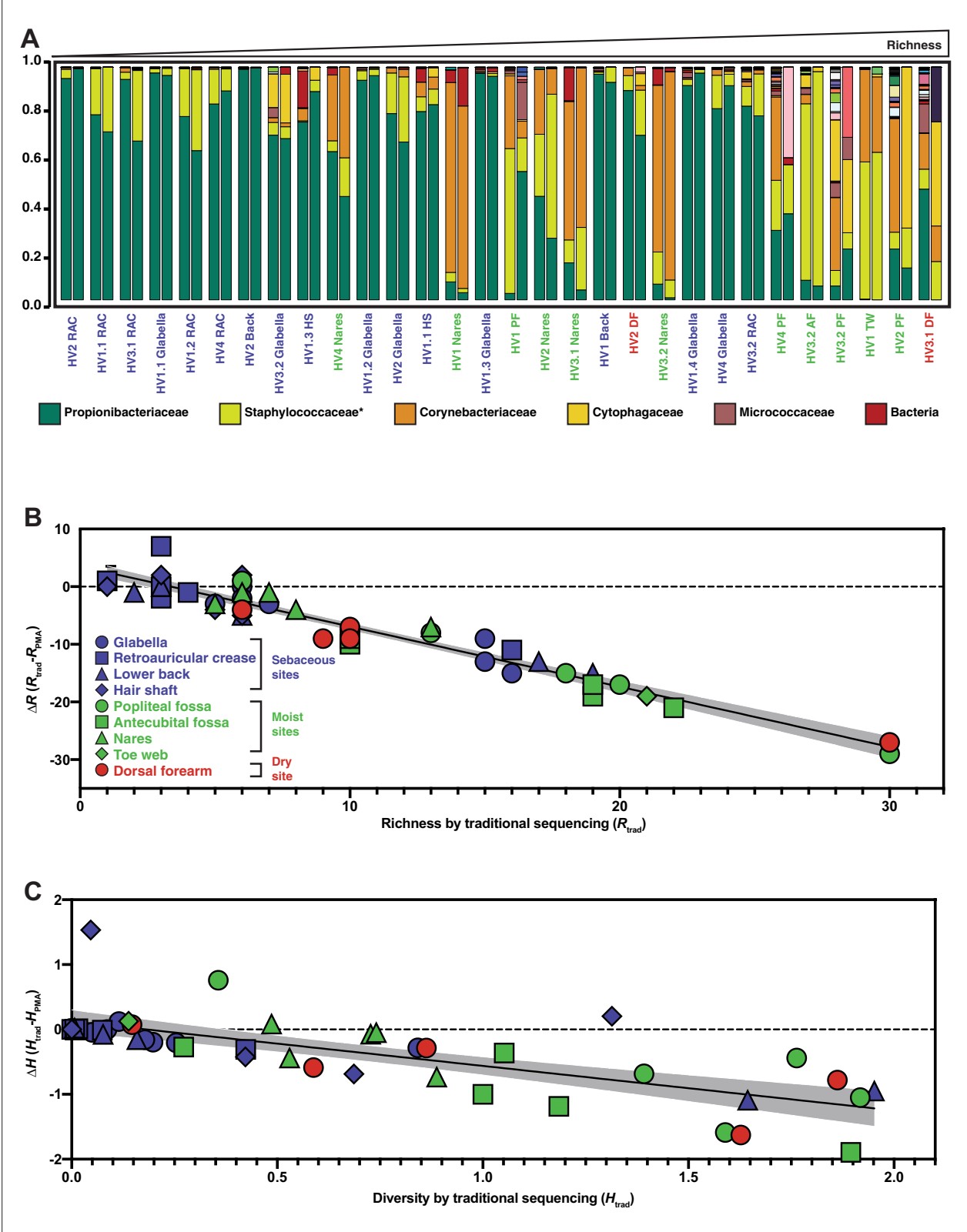

**Figure 3.** Relative abundance and change in richness and diversity of traditional sequencing compared to PMA-seq. (**A**) Relative abundance of all sequenced bacterial taxa at the family level. Paired bars represent data from traditional sequencing (left) and PMA-seq (right). Samples are ordered by increasing richness in traditional sequencing. Labels below each pair of bars indicate each sample's donor, replicate, and site (e.g., HV1.1 RAC indicates Healthy Volunteer 1, replicate sample 1, retroauricular crease). HS: hair shaft; RAC: retroauricular crease; VF: volar forearm; PF: popliteal fossa; TW: toe

*Figure 3 continued on next page*

*Figure 3 continued*

web; AF: antecubital fossa. *Relative abundance data for Staphylococcaceae was determined using forward-read sequencing information only. Samples with fewer sequencing reads than PBS controls are not displayed. All identified bacterial taxa with corresponding colors can be found in *Figure 3— figure supplement 1*. (**B**) The richness changes between traditional sequencing ($R_{trad}$) and PMA-seq ($R_{PMA}$) are demonstrated by plotting the change in richness ($\Delta R$) against $R_{trad}$. Colors represent different site types and shapes represent different sample sites. The shaded gray region represents the 95% confidence interval for the linear regression. (**C**) The Shannon diversity changes between traditional sequencing ($H_{trad}$) and PMA-seq ($H_{PMA}$) are demonstrated by plotting the change in diversity ($\Delta H$) against $H_{trad}$.

The online version of this article includes the following figure supplement(s) for figure 3:

**Figure supplement 1.** Full list of identified taxa with corresponding colors.

different taxa), in reality the richness across the skin microbiome at different body sites is relatively similar and low (1–10 different taxa). Thus, though there appears to be a wide range of diversity in the skin microbiome by traditional sequencing, PMA-seq indicates that this is generally an overestimation at any given skin site. In fact, our results suggest that the viable skin microbiome tends to be dominated by a relatively small number of taxa at most sites. An independent analysis of glabella samples from five healthy volunteers including contamination controls revealed the same trends that traditional sequencing overrepresented both diversity and richness with or without contaminant removal (*Figure 6—figure supplement 3*).

## Most bacterial groups are overrepresented by traditional sequencing methods

The metrics of richness and diversity offer important information regarding how the composition of the skin microbiome changes at different body sites between traditional sequencing and PMA-seq. We further wanted to understand the changes in the relative abundance of specific bacterial taxa. To quantify taxon-level PMA-dependent changes, we developed a PMA index ($I_{PMA}$) for each bacterial taxon, which is calculated as follows: $I_{PMA} = \frac{A_{PMA}}{(A|PMA) + (A|trad)}$, where $A_{PMA}$ is the relative abundance by PMA-seq and $A_{trad}$ is the relative abundance by traditional sequencing. A low PMA index (<0.5) indicates that the taxon in question is overrepresented by traditional sequencing, while a high PMA index (>0.5) indicates that the taxon in question is underrepresented by traditional sequencing (*Figure 4A*). We note that these values represent enrichment relative to the rest of the sequences, a measurement that is distinct from the viability score. Calculating PMA indices revealed that the abundances of most bacterial taxa at any given body site are overestimated by traditional sequencing, as most taxa had PMA index values close to 0 (*Figure 4A*). 16S rRNA copy numbers could affect these values but the 16S copy numbers of the most abundant species were all within a roughly twofold range, such that this effect is minor compared to the trends observed (*Stoddard et al., 2015*).

The four most abundant bacterial families (*Propionibacteriaceae, Staphylococcaceae, Corynebacteriaceae*, and *Micrococcaceae*) made up 93% of total sequencing reads (96% of PMA-seq reads and 91% of traditional sequencing reads) and demonstrated interesting family-level PMA index patterns. The family *Propionibacteriaceae* includes a major component of the skin microbiome, *C. acnes*, which has been shown by traditional sequencing to comprise upwards of 50% of the skin microbiome irrespective of site type (*Byrd et al., 2018*). PMA-seq revealed that traditional sequencing accurately represents *Propionibacteriaceae* abundance in sebaceous sites (demonstrated by a PMA index close to 0.5), but overrepresents *Propionibacteriaceae* in moist and dry sites (PMA indices of 0.2–0.3). Furthermore, *Propionibacteriaceae* dominated sebaceous sites (accounting for >75% of all viable bacteria in most sebaceous samples), but did not dominate moist or dry sites (their viable abundance did not exceed 50% of all viable bacteria in any of those samples) (*Figure 4A and B*). Similar to the *Propionibacteriaceae*, bacteria in the family *Staphylococcaceae* appeared to be well-represented across all body sites (*Figure 4A and B* and *Figure 4—figure supplement 1*).

Bacteria in the family *Corynebacteriaceae* are also considered main constituents of the skin microbiome, but our results showed that traditional sequencing overestimates the abundance of *Corynebacteriaceae* at every skin site except for the nares. For example, traditional sequencing identified a high abundance of *Corynebacteriaceae* in the popliteal fossa, but PMA-seq showed that these reads were largely of inviable origin (*Figure 4B*). Previous studies have demonstrated that *Corynebacteria*

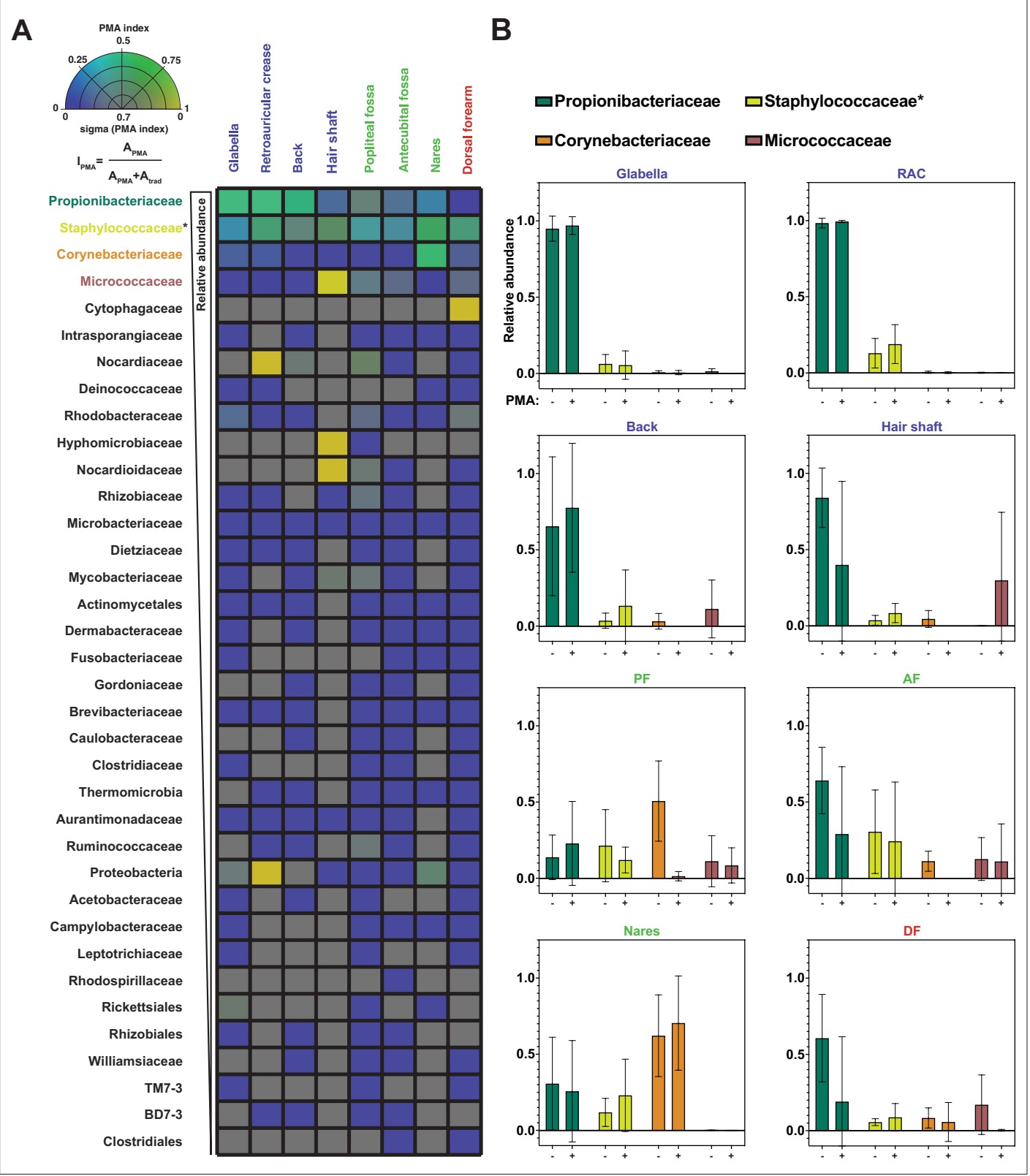

**Figure 4.** Propidium monoazide (PMA) index and change in relative abundance between traditional and PMA-seq. (**A**) The PMA index for each bacterial taxon that was present in at least four samples is shown here as an average between samples of the same sample site shown in *Figure 3*. Color indicates PMA index value. Saturation indicates confidence (sigma) in the PMA index value and was calculated using the standard deviation of PMA index across the samples that went into that pixel. Bacterial taxa are ordered by decreasing overall relative abundance. Each square represents the average of at

*Figure 4 continued on next page*

*Figure 4 continued*

least four samples taken from different individuals. PMA index is calculated by comparing the relative abundance of a given taxon as measured by PMA-seq ($A_{PMA}$) to the sum of the relative abundance for that taxon in both traditional sequencing ($A_{trad}$) and PMA-seq. (**B**) Relative abundance at each body site for the top three most abundant (overall) bacterial taxa as assessed by traditional sequencing and PMA-seq. Colors of bars correspond to colors in *Figure 2F*. Error bars represent standard deviation. *Relative abundance data for Staphylococcaceae was determined using forward-read sequencing information only.

The online version of this article includes the following figure supplement(s) for figure 4:

**Figure supplement 1.** Viability score for three skin sites using lysostaphin and *Staphylococcus*-specific PCR primers.

are readily cultured from nasal isolates, which supports our PMA-seq finding that viable members of this taxon are abundant in the nares but not at most other skin sites (*Liu et al., 2020*).

Interestingly, *Micrococcaceae* were overrepresented by traditional sequencing at every site except for the hair shaft. In the hair shaft, *Micrococcaceae* were abundant by PMA-seq but almost undetectable by traditional sequencing. As shown in *Figure 4B*, the increase in the relative proportion of viable *Micrococcaceae* detected by PMA-seq corresponds to a decrease in viable *Propionibacteriaceae*, suggesting that *Micrococcaceae* may not be detected by traditional sequencing because of the high abundance of DNA from inviable *Propionibacteriaceae*. These results suggest that most skin sites are colonized by a relatively small number of bacterial families, different families distinctly colonize different skin sites, and the majority of the additional bacterial DNA on the skin surface come from inviable bacteria.

## The spatial organization and viability of mouse skin microbiome are similar to that of humans

While our human skin microbiome FISH and PMA studies strongly agree, we are unable to directly compare the two approaches on the same samples. For a system in which we could perform both FISH and PMA-ddPCR on the same samples, we turned to mouse skin, which also enabled us to ask whether our findings are human-specific. We assessed the spatial distribution of bacterial cells in mouse skin tissue using the universal bacterial EUB338 FISH probe with tissue from K14-H2B-GFP mice. We observed the same bacterial distributions as seen in the human tissues: a high abundance of bacteria in hair follicles (enrichment score of 15.26) with relatively few bacteria on the skin surface (enrichment score of 0.21) (*Figure 5A and D*). To test how fur impacts the presence of bacteria on the skin surface, we performed FISH staining on skin from nude mice (SKH1-Hrhr Elite) and found similar bacterial distributions (follicle-associated enrichment score of 10.79 compared to 1.13 for the skin surface) (*Figure 5B and D*). The significant numbers of bacteria observed in hair follicles suggest that the absence of surface bacteria is not merely due to the relatively clean conditions in which laboratory mice are housed. Because the mouse tissue was untreated with any sterilizing agents prior to dissection, these results also suggest that the low numbers of skin-surface-associated bacteria found in the human tissue samples are not simply a result of pre-biopsy sterilization.

Since we did not detect many bacteria on the surface of any of the skin samples tested, we sought a positive control to confirm that our FISH staining can visualize bacteria on the skin surface if they are present. For example, it is possible that the dry, acidic nature of the skin affects FISH efficacy, leading to little staining at the skin surface. To this end, we applied *E. coli* cells to dorsal mouse skin tissue after removing it from the animal. This tissue was then processed in the same way as the human and other mouse tissue. FISH staining revealed many bacteria on the surface of these samples, confirming that this technique can be used to reliably visualize bacteria on the skin surface (*Figure 5C*). As a negative control, we also confirmed that a probe encoding the reverse complement of the EUB338 FISH probe (NONEUB338) did not significantly hybridize to the skin surface or follicles (*Figure 5C*). These results support the conclusions from our FISH experiments on human tissues by demonstrating EUB338 staining works well in the biological context of the skin surface.

We next assessed the viability of bacteria in the mouse skin microbiome using PMA-ddPCR. As further evidence that our previous findings are not human-specific, the PMA-ddPCR-based viability score for mouse skin microbiome sites was similar to the average viability score for human skin sites (0.066 and 0.045, respectively) and was much lower than the viability score for the mouse or human fecal microbiome (0.98 and 0.66, respectively) (*Figure 5E*). These results indicate that, despite having

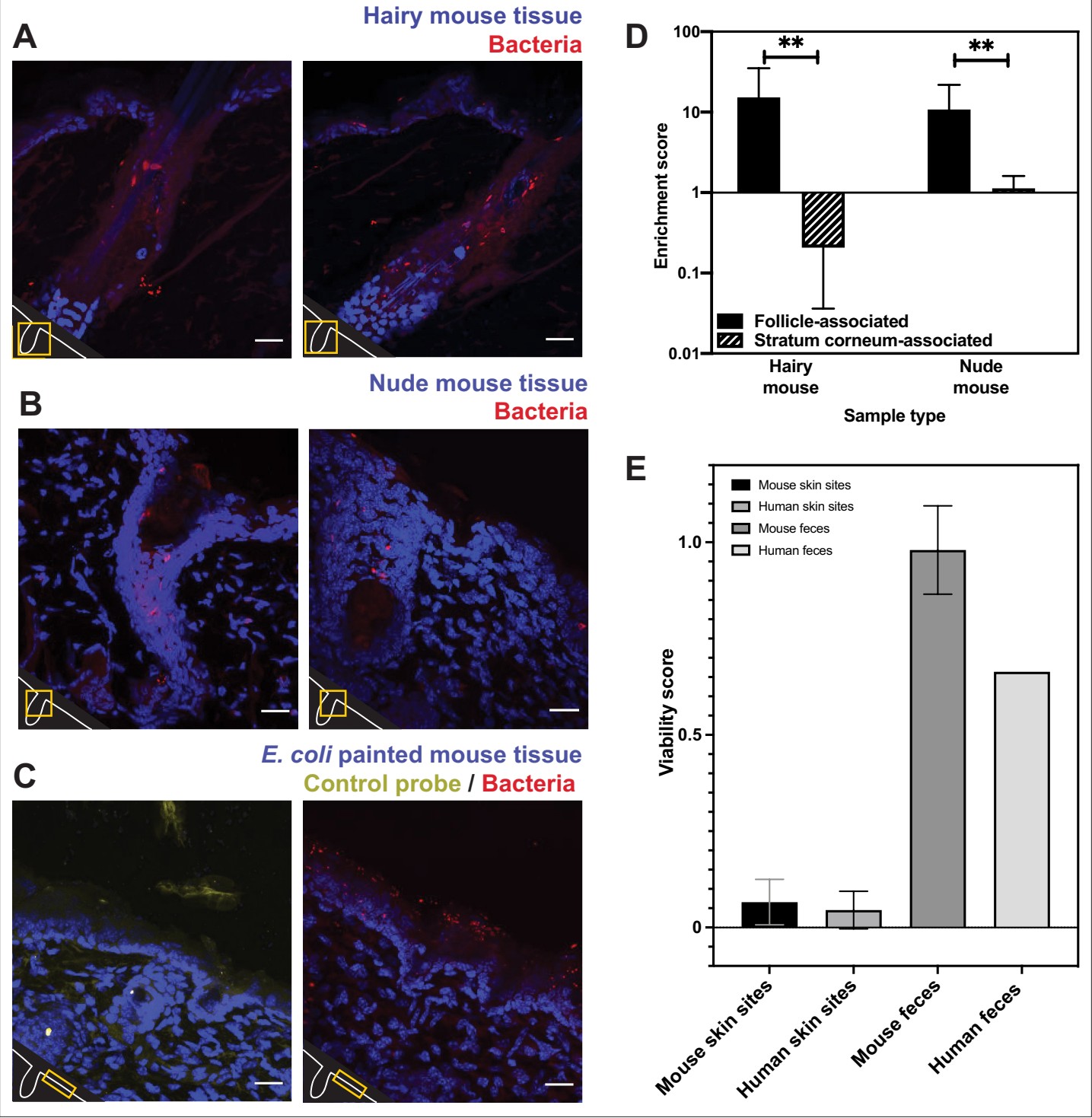

**Figure 5.** Bacterial fluorescence in situ hybridization (FISH) staining of mouse tissue (**A–D**) and comparison of mouse viability scores and human viability scores. (**A**). Tissues from a K14-H2B-GFP mouse stained with EUB338 show abundant bacterial signal in hair follicles but not on the skin surface. (**B**) Tissues from SKH1-Hrhr Elite nude mice also show bacterial presence concentrated to cutaneous structures and not at the skin surface. (**C**) *E. coli* applied to C57BL/6 mouse tissue was stained with either EUB338 (in red) or its complementary strand control probe NONEUB338 (in yellow). (**D**) Quantification enrichment scores showing the median and interquartile range. Significance was calculated using the Mann–Whitney test. *p≤0.05, **p≤0.01, N = 6 for hairy mouse follicle, nude mouse follicle, and hairy mouse stratum corneum, N = 5 nude mouse stratum corneum. (**E**) The PMA-ddPCR-based viability scores for mouse skin microbiomes are much lower than for mouse fecal microbiomes (0.66 and 0.98, respectively). These viability scores for mouse sites are very similar to those for humans (0.066 and 0.045 for skin microbiomes, 0.98 and 0.66 for fecal microbiomes). Error bars represent standard deviation.

distinct skin biology, both humans and mice have an abundance of bacterial DNA on the skin surface that is not associated with viable cells.

## Skin surface repopulation is driven by stable communities of bacteria below the skin surface

Finally, we sought to determine the functional significance of our findings in the context of skin microbiome surface perturbation and repopulation. Specifically, we performed a perturbation-recovery experiment in which the skin microbiome of the forehead from healthy volunteers was sampled at time T = 0 to establish a baseline microbial community, sterilized with benzalkonium chloride, and then sampled 3, 24, and 48 hr later to monitor recovery (*Figure 6A*). At each sampling timepoint, half of the sample was treated with PMA prior to DNA isolation and subsequent 16S rRNA gene sequencing. Focusing on the PMA-untreated samples, we found that four of the five volunteers demonstrated significant reduction in the absolute amount of DNA upon sterilization (*Figure 6B*), but that the overall bacterial composition of the skin microbiome remained relatively stable (*Figure 6A*, all bacteria listed in *Figure 6—figure supplement 2*.). It should be noted that, although the skin microbiome composition found in volunteer 4 is markedly different than the communities identified in the other volunteers, the species identified are still common constituents of a healthy skin microbiome. Additionally, volunteer 4 had a lower abundance of bacterial DNA overall, which might lead to the lower relative abundance of *Propionibacteriaceae*.

Our findings recapitulate the previously reported paradox that the skin microbiome is both unstable at short timescales and stable at long timescales (*Nielsen and Jiang, 2019*; *Oh et al., 2016*). Importantly, analysis of the PMA-treated samples enabled us to understand the recovery dynamics. Specifically, we used Bray–Curtis dissimilarity to monitor the extent of population recovery (*Figure 6C*). We found that comparing samples with and without the use of PMA to the PMA-treated baseline community resulted in a consistent pattern across all individuals. Over the course of the experiment, the communities converged back to the PMA-treated baseline samples but showed no consistent patterns of similarity relative to the PMA-untreated baseline samples (*Figure 6—figure supplement 2A*). This result supports the hypothesis that bacterial DNA on the skin surface can be readily removed and is continuously replenished by viable bacterial populations in protected reservoirs below the skin surface.

Examining repopulation dynamics more finely at the ASV level revealed that ASVs that were lost upon surface sterilization often reappeared over time, consistent with the model that surface repopulation can be driven by subsurface populations. Specifically, we found that after surface sterilization, the skin is repopulated by the same ASVs as were present before sterilization (*Figure 6—figure supplement 2B*). In *Figure 6—figure supplement 2B*, we highlight specific examples in which a given ASV appears in the live-cell population (+PMA) prior to surface sterilization (T0), disappears from the live-cell population following sterilization (T3), and later reappears in the live-cell population either 24 or 48 hr later. Together, these results suggest that bacteria thrive in protected areas below the skin surface and that the DNA of their dead remains accumulates on the skin surface. In this way, the bacterial DNA on the skin surface acts as a fingerprint of the communities below.

## Discussion

Here, we used both imaging and PMA-based methods to demonstrate that the skin surface is sparsely colonized by bacteria. This central finding holds true across skin from human biopsies, healthy swabbed volunteers, hairy mice, and nude mice. The skin microbiome has garnered a great deal of attention as a means for educating the immune system, combatting pathogens, and promoting wound healing, and multiple groups are pursuing skin probiotics (*Yu et al., 2020*). Our findings have significant implications for the mechanisms underlying these skin microbiome functions, as well as for the ability to manipulate skin microbiome composition. For example, our findings support previous work suggesting that the key function of immune education by the skin microbiome occurs within hair follicles (*Polak-Witka et al., 2020*), though cell-free DNA on the skin surface could also impact the immune system. Our findings also support previous work showing that individual pores are colonized by clonal bacterial populations (*Conwill et al., 2022*). Meanwhile, our results may help to explain why stably colonizing the skin surface with exogenous bacteria has proved to be difficult and often requires

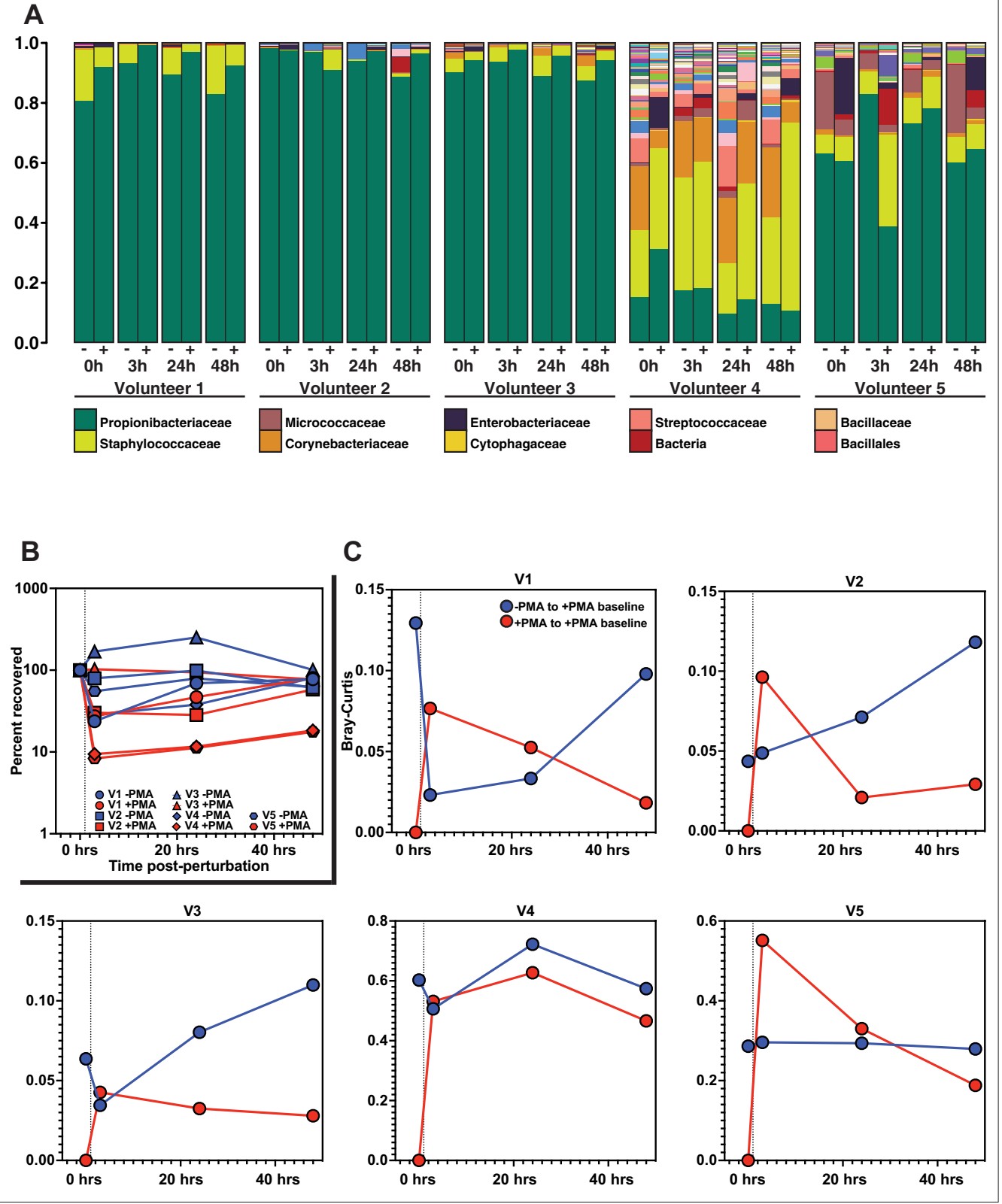

**Figure 6.** Skin microbiome perturbation and recovery. (**A**) Bacterial relative abundance in each individual over the 48 hr following perturbation. 0 hr represents baseline, pre-perturbation community. Whether a sample was treated with propidium monoazide (PMA) is indicated by (-) and (+). (**B**) Quantification of bacterial DNA recovery over the 48 hr following perturbation. DNA was quantified using droplet digital PCR (ddPCR). (**C**) Bray–Curtis dissimilarity of each individual over the 48 hr following perturbation. Red data points are comparing PMA-treated samples to the PMA-treated

*Figure 6 continued on next page*

*Figure 6 continued*

baseline sample. Blue data points are comparing PMA-untreated samples to the PMA-treated baseline sample. Dashed vertical line indicates the point of perturbation.

The online version of this article includes the following figure supplement(s) for figure 6:

**Figure supplement 1.** List of bacteria identified in perturbation recovery.

**Figure supplement 2.** Skin microbiome perturbation and recovery.

**Figure supplement 3.** Contamination removal performed on 600 nt sequencing data.

abrasion (*Scharschmidt et al., 2015*). Disruption of the stratum corneum (skin surface layers), which occurs during skin abrasion, may allow access to the deeper layers of tissue where the stable bacterial populations reside. Thus, targeting the bacteria within hair follicles may represent a better strategy for stably manipulating the skin microbiome or educating the immune system. While it has been known for some time that bacteria inhabit hair follicles (*Polak-Witka et al., 2020*; *Lousada et al., 2021*), our findings extend this knowledge and suggest that the viable bacteria of the skin microbiome are primarily restricted to these sites. This is further supported by our finding that bacteria can be readily cultured from the skin surface, but at far less abundance than suggested by the amount of bacterial DNA present on the skin.

Our results also offer insight into the apparent paradox of skin microbiome stability, in which skin microbiome composition has been found to be both stable over long timescales and susceptible to transient perturbations (*Nielsen and Jiang, 2019*; *Oh et al., 2016*). The skin is primarily colonized by viable bacteria in pilosebaceous units and other skin invaginations, and the bacterial DNA on the skin surface is mostly from dead bacteria. We show that surface bacterial DNA could be easily washed away, while the viable subsurface bacteria remain unperturbed by washing, persisting as a small but stable source of new bacterial DNA that replenishes the skin surface over time. This model also explains the counter-intuitive abundance of obligately anaerobic bacterial species associated with the skin microbiome. Human skin cells are also born below the skin surface and are pushed to the surface as they die. In this way, the life cycle of the skin microbiome may resemble the process of epithelial cell turnover in the skin.

Our findings also raise fundamental questions that will need to be addressed by future studies focused on why the skin surface is poorly colonized. The accessible DNA in non-skin microbiomes is generally representative of viable bacterial cells as all non-skin microbiomes tested had viability scores above 0.4. This is in stark contrast to the skin microbiome, which had viability scores between 0.02 and 0.12. Even saliva, which contains many antimicrobial compounds (*van 't Hof et al., 2014*), had a viability score nearly four times greater than the highest viability score for a skin microbiome. The uniquely low viability score associated with the skin could be explained by passive mechanisms, like bacterial DNA adhering to the skin long after a bacterial cell dies. Alternatively, there could be active mechanisms like bacterial killing on the skin surface by factors like antimicrobial peptides produced by epithelial cells, competition between bacterial species, or exposure to harsh environmental factors such as starvation, UV radiation, or desiccation. Another open question is whether similar trends hold true for non-bacterial components of the skin microbiome like fungi and viruses.

## Materials and methods

### Human and mouse tissue processing

Human tissue was obtained through the Skin Translational Research Core within the Skin Biology and Diseases Resource-based Center (SBDRC) at the University of Pennsylvania. Normal adult human skin was obtained during skin cancer surgery procedures. After the cancerous tissue is removed, normal (non-cancerous) tissue must also be removed to ensure a cosmetic repair. These tissue portions are normally discarded, but were collected for imaging. Surgical scrub solution was used prior to the procedure (alcohol/iodine or chlorhexidine). Tissues were embedded in room temperature OCT immediately upon excision and stored at –80°C. Tissues were obtained from the forehead of a 32-year-old female and the cheek of an 81-year-old male. Tissues were obtained through the Core B- Skin Translational Research Core at the SBDRC as part of the 'Fresh normal and diseased human skin' service.

All mice were housed and maintained in a certified animal facility, and all experiments were conducted according to USA Public Health Service Policy of Humane Care and Use of Laboratory Animals. All protocols were approved by the Institutional Animal Care and Use Committee, protocol #1867 (Princeton University). Dorsal skin from K14-H2B-GFP, C57BL/6J, and SKH1-Elite nude mice was used for fluorescence experiments. The fur from K14-H2B-GFP and C57BL/6J mice was shaved using clippers prior to dissection. Dorsal tissue was removed, cut into thin (~5 mm) strips, and embedded in room temperature OCT. OCT blocks were placed on dry ice to freeze and subsequently stored at –80°C. Both sexes were used. All animals were between 3 and 6 wk of age and were fed standard, non-sterilized rodent chow. No pathogens were identified in the regular health screenings conducted by facility staff. Mice were housed with 1–5 animals per cage with bedding and enrichment using Ventilated Thoren racks. Mice were euthanized using $CO_2$ inhalation (as per the AVMA Guidelines for the Euthanasia of Animals: 2020 Edition), followed by cervical dislocation. Dissections were performed with ethanol-sterilized tools.

## Fluorescence in situ hybridization and imaging

Human and mouse tissues were processed identically. 30 µm tissue sections were sectioned and mounted on slides using a Leica CM3050S cryostat. Tissues on slides were fixed in 4% formaldehyde in 1× PBS for 20 min. Following fixation, tissues were washed for 5 min in 1× PBS and then incubated in hybridization buffer (0.9 M NaCl, 20 mM Tris-HCl, 10% formamide) containing 10 mg/mL lysozyme and FISH probes at 1 µg/µL for 2–3 hr at 47°C in the dark. Nucleotide sequences of FISH probes can be found in *Supplementary file 1*. After hybridization, slides were incubated in wash buffer (0.9 M NaCl, 20 mM Tris-HCl) for 1 hr at 47°C in the dark. Slides were then washed in 1× PBS for 10 mins. To visualize tissue nuclei, tissues were then stained with 1 µg/mL DAPI for 10 min at room temperature. Slides were then washed 3× in 1× PBS for 10 min each. Following the final wash step, tissues were mounted with glycerol-based anti-fade non-curing mounting media. Coverslips were sealed using a 1:1:1 mixture of petroleum jelly, lanolin, and paraffin. Images were acquired on a Nikon A1R-Si HD confocal microscope controlled by NIS Elements software. ImageJ and MATLAB (The MathWorks, Natwick, MA) were used for image processing. To calculate fluorescence enrichment scores, maximum projections of Z-stacks were used. A rectangular region of interest (ROI) of width W was drawn around a follicle using only the DAPI channel. A second ROI was drawn near the follicle opening with dimensions 0.5W × 2W in order to capture follicle-associated fluorescence near the follicle opening. In an orthogonal processing step, the pixel intensities for the entire image in the FISH fluorescent channel were fit using a 3 component Gaussian mixture model (GMM) corresponding roughly to non-tissue background, non-probe autofluorescence, and probe-based signal. Using this GMM as the background subtraction value for each image, the mean intensity for each ROI was calculated. This threshold value was used to calculate the mean fluorescence value inside the ROI and outside the ROI. The ratio of the mean fluorescence value inside the ROI to the mean fluorescence value outside of the ROI was used for quantifying follicle- and stratum corneum-associated fluorescence (enrichment score). Significance was calculated using the Mann–Whitney test.

Stationary phase cultures were grown according to the following conditions:

| Strain | Growth time (hours) | Media | Environmental condition |
|---|---|---|---|
| *Cutibacterium acnes* ATCC6919 | 72 | Reinforced clostridial medium (RCM) (Oxoid, CM0149) | Anaerobic, 37°C |
| *Corynebacterium striatum* ATCC6940 | 24 | Brain heart infusion (BHI) (BD, 237500) | Aerobic, 30°C |
| *Micrococcus luteus* ATCC4698 | 24 | Tryptic soy broth (TSB) (BD, 211825) | Aerobic, 30°C |
| *Staphylococcus epidermidis* EGM 2-06 | 24 | Tryptic soy broth (TSB) (BD, 211825) | Aerobic, 37°C |

## Human subject microbiome samples

Microbiome samples were collected under Princeton University IRB #13003 at the Princeton University Department of Molecular Biology. Healthy volunteers were recruited via informational flyers.

Volunteers gave informed consent prior to sample collection. Participants were healthy volunteers, male and female, white, aged 26–35 y, with no history of chronic skin conditions or autoimmune disease, and were not using antibiotics at the time of sampling or 1 mo prior to sample collection.

Skin microbiomes from healthy volunteers were collected using sterile foam-tipped collection swabs pre-moistened with sterile 1× DPBS. Though often included, we chose not to use detergent in the swabbing buffer in order to avoid negatively affecting bacterial cell membranes and altering viability scores. *Figure 2—figure supplement 1G* shows how swabbing with and without 0.1% Triton X-100 (Sigma) affects viability scores, ddPCR results, and CFU. Areas of interest were sampled for 60 s before being resuspended in sterile 1× DPBS. Tongue microbiome samples were collected using sterile foam-tipped collection swabs. Hair shaft samples were collected by plucking hairs and using only the bulb portion. Follicle contents were collected using Bioré Deep Cleansing Pore Strips (Kao USA Inc, Cincinnati, OH). Saliva was collected in sterile 50 mL conicals from healthy volunteers. Plaque was collected by scraping the teeth of healthy volunteers using sterile toothpicks and resuspending the collection in sterile 1× DPBS. Murine fecal samples from C57BL/6 mice were collected during dissection. Human skin microbiome samples shown in *Figure 2—figure supplement 1* were plated for CFU calculations prior to the addition of PMA. Samples were plated on blood agar plates (5% sheep blood in tryptic soy agar, VWR International) and grown for 24–48 hr aerobically or anaerobically. For perturbation recovery experiments, volunteers were asked to refrain from using any type of products on their face for the duration of the experiment (48 hr) as well as 24 hr prior to baseline sampling. Volunteers were allowed to bathe/shower, but were asked to refrain from washing their faces. Skin surface sterilization was done by scrubbing the sample area for 60 s with an antiseptic wipe containing benzalkonium chloride 0.13% w/v (North by Honeywell, #150910). To ensure that the process of sampling the skin surface would not affect the bacterial repopulation dynamics, forehead of each volunteer into four sections of equal areas: T0 (baseline, before sterilization), T1 (3 hr post sterilization), T2 (24 hr post-sterilization), and T3 (48 hr post-sterilization). Each section was sampled for 60 s using a sterile foam-tipped collection swab pre-moistened with sterile 1× DPBS. As described above, each swab was then resuspended in sterile 1× DPBS.

The human fecal sample was collected by the Donia lab under Princeton University IRB #11606 and was gifted to the Gitai lab. This sample and the methods of collection were described in *Javdan et al., 2020*.

## Heat-killed *E. coli* ratios

In order to demonstrate the efficacy of PMA, known ratios of live and heat-killed *E. coli* cultures were mixed and subjected to PMA treatment. First, an overnight culture of *E. coli* NCM3722 was back-diluted into fresh LB media at a ratio of 1:1000 and grown at 37°C for 4 hr to reach mid-log. Stationary-phase experiments were done with overnight, 18 hr cultures. Cultures were then washed 3× and suspended in sterile PBS. Half of the culture was heat-killed by incubating at 70°C for 20 min while the other half remained at room temperature. The heat-killed *E. coli* cultures were allowed to cool to room temperature before combining with non-heat-killed *E. coli* cultures to achieve a 50% (by volume) heat-killed mixture. For the 0% heat-killed, no heat-killed bacteria were added. Likewise, for the 100% heat-killed, only heat-killed bacteria were used. Each condition was mixed well and then split evenly between two sterile 1.5 mL microcentrifuge tubes.

## PMA treatment and DNA isolation

After collection, samples were split evenly between two sterile 1.5 mL microcentrifuge tubes. PMA (Biotium Inc) was added to one of the two tubes to a final concentration of 50 μM. All tubes were incubated in the dark at room temperature for 10 min before being exposed to light to cross-link PMA molecules using the PMA-Lite LED Photolysis Device (Biotium Inc). DNA was then isolated from all samples using either the DNeasy PowerSoil Kit (QIAGEN) or according to the protocol outlined in *Meisel et al., 2016*. The Meisel et al. protocol was modified such that sonication (Branson Sonifier 250, VWR Scientific) was used instead of bead-beating for the purposes of cell lysis. Both DNA isolation methods work similarly (*Figure 2—figure supplement 1H*). If lysostaphin (Sigma-Aldrich) was used, it was added following PMA activation and before DNA isolation to a final concentration of 0.1 mg/mL and incubated at room temperature for 30 mins.

## Droplet digital PCR (ddPCR)

The Bio-Rad QX200 AutoDG Droplet Digital PCR System was used to quantify extracted DNA from microbiome samples and from pure bacterial cultures. Reaction mixtures contained 2× QX200 ddPCR EvaGreen Supermix and universal 16S qPCR primers at 10 nM concentrations in a total volume of 25 μL. Primer sequences can be found in *Supplementary file 1*. Reaction mixtures were transferred to sterile ddPCR 96-well plates (Bio-Rad #12001925) which were loaded into the QX200 Automated Droplet Generator. After droplet generation, the plate was heat-sealed using the PX1 PCR Plate Sealer (Bio-Rad #1814000) and PCR was performed with a pre-step of 95°C for 5 min followed by 40 rounds of amplification with 60°C, 1 min extensions and a final hold temperature of 12°C using a C1000 Touch Thermal Cycler (Bio-Rad #1851197). Samples were subsequently loaded into the QX200 Droplet Reader for quantification. Automatic thresholding was performed using the Quantasoft software and subsequently exported to Microsoft Excel for analysis. Significance was calculated using a Student's *t*-test. To calculate the viability score for a given pair of '-PMA' and '+PMA' matched samples, the following calculation was done:

$$\frac{Copies\ per\ 20\ \mu L\ with\ PMA}{Copies\ per\ 20\ \mu L\ without\ PMA}$$

## ddPCR and CFU standard curves

Cultures of *S. epidermidis* EGM 2-06 were grown in tryptic soy broth (TSB) overnight and diluted 1:1000 the following morning in TSB and grown for 4 hr until a final OD of 0.4. Tenfold dilutions of *S. epidermidis* culture were then prepared, plated for CFU on 5% sheep blood in tryptic soy agar (VWR International), and divided between two 1.5 mL microcentrifuge tubes. Stationary-phase cultures of *C. acnes, S. epidermidis, M. luteus,* and *C. striatum* were grown according to the table above. PMA was added to one tube for a final concentration of 50 μM and the other tube was left untreated. PMA activation and DNA isolation was then done according to the methods outlined above.

## 16S rRNA gene amplicon sequencing

DNA was isolated from microbiome samples (with and without PMA) using the DNeasy PowerSoil Kit (QIAGEN) (*Figures 2–5*) or according to the protocol outlined in Meisel et al. (*Figure 6*). The V1-V3 region of the 16S gene was amplified using the primers 27F (5'- AGAGTTTGATCCTGGCTCAG ) and 534R (5'- ATTACCGCGGCTGCTGG). Illumina sequencing libraries were prepared using previously published primers (*Caporaso et al., 2012*). Libraries were then pooled at equimolar ratios and sequenced on an Illumina MiSeq Micro 500 nt (*Figures 3 and 4*) or MiSeq V3 600 nt (*Figure 6*) as paired-end reads. Reads were 2 × 250 bp or 2 × 300 bp with an average depth of ~33,616 reads for 500 nt and 245,488 reads for 600 nt (*Supplementary files 2 and 3*). Also included were 8 bp index reads, following the manufacturer's protocol (Illumina, USA). Raw sequencing reads were filtered by Illumina HiSeq Control Software to generate Pass-Filter reads for further analysis. Index reads were used for sample de-multiplexing. Amplicon sequencing variants (ASVs) were then inferred from the sequences using the DADA2 plugin within QIIME2 version 2018.6 (*Bolyen et al., 2018*; *Callahan et al., 2016*). Reverse reads were trimmed to 245 bp for the 2 × 250 bp data, and the reverse reads were trimmed to 275 bp for the 2 × 300 bp data. For the 2 × 250 bp data, two different ASV inference methods were utilized: one using only the forward reads (to capture the longer-than-average *Staphylococcus* amplicons) and one using both paired-end reads. For the 2 × 300 bp data, ASV inference was performed using both paired-end reads. ASV inference using forward reads was performed using DADA2's denoise-single function,while ASV inference using both reads was performed using the denoise-paired function. Taxonomy was assigned to the resulting ASVs with a naive Bayes classifier trained on the GreenGenes database version using only the regions of the 16S rRNA gene spanned by the ASVs (*Bokulich et al., 2018*; *McDonald et al., 2012*). All downstream analyses were performed using family-level taxonomy assignments unless specified to be at the ASV level. PBS control samples were included at the library preparation step for 500 nt sequencing data, and air swab DNA isolation controls were included for 600 nt sequencing data. Contamination removal was performed on 600 nt sequencing data using the Decontam package for R (*Figure 6* and *Figure 6—figure supplement 2* show data without contamination removal; *Davis et al., 2018*). *Figure 6—figure supplement 3* shows how Shannon diversity and richness change between traditional sequencing and PMA-seq (similar to what is shown in *Figure 3B and C*). Removal of contaminants (using either 0.1 or 0.2 threshold)

has little effect on the data, suggesting that, although contamination removal cannot be performed on the 500 nt sequencing data due to the absence of air swab controls, this should not change the overall findings shown in *Figures 3 and 4*. Relative abundance, richness, Shannon diversity, Bray–Curtis dissimilarity, and PMA -index were assessed using the Vegan package for R or Microsoft Excel and plotted using R, Prism, and MATLAB (*J. Oksanen, F. G. Blanchet, M. Friendly, R. Kindt, P. Legendre, D. Mcglinn, P. R. Minchin, R. B. O'Hara, G. L. Simpson, P. Solymos, M. H. H. Stevens, E. Szoecs, H, 2019*). The PMA index was calculated using relative abundance and was not calculated for any bacterial taxa that was present in fewer than four samples.

## Acknowledgements

We thank all members of the Gitai lab for their insights and comments. We also thank Dr. Gary Laevsky and the Princeton Molecular Biology Microscopy Core, which is a Nikon Center of Excellence, for microscopy support; Dr. Wei Wang and the Genomics Core Facility in The Lewis Sigler Institute for Integrative Genomics at Princeton University for support with 16S rRNA gene amplicon sequencing; Matthew Cahn for his assistance with processing 16S rRNA gene amplicon sequencing data; Bahar Javdan for processing the human fecal sample; Elizabeth Grice for support and feedback; and Laurice Flowers and Elizabeth Grice for providing the *Staphylococcus epidermidis* strain used. Funding was provided in part by NIH (DP1AI124669 to ZG, EMA, and BPB, and T32 GM007388 to EMA). Additional funding provided by the National Science Foundation (NSF PHY-1734030 to BPB). Research reported in this publication was also supported by the National Center for Advancing Translational Sciences of the National Institutes of Health under Award Number TL1TR003019 (EMA) and by the Schmidt Transformative Technology Fund (EMA). This work was supported in part by the Penn Skin Biology and Diseases Resource-based Center (P30-AR068589) and the University of Pennsylvania Perelman School of Medicine. The contents are solely the responsibility of the authors and do not necessarily represent the official views of the respective funding agencies.

## Additional information

### Competing interests

Danelle Devenport: Reviewing editor, *eLife*. The other authors declare that no competing interests exist.

### Funding

| Funder | Grant reference number | Author |
| --- | --- | --- |
| National Institute of Allergy and Infectious Diseases | DP1AI124669 | Ellen M Acosta<br>Benjamin P Bratton<br>Zemer Gitai |
| National Institute of General Medical Sciences | GM007388 | Ellen M Acosta |
| National Science Foundation | PHY-1734030 | Benjamin P Bratton |
| National Center for Advancing Translational Sciences | TL1TR003019 | Ellen M Acosta |
| Penn Skin Biology and Diseases Resource-based Center | P30-AR069589 | Aimee S Payne |

The funders had no role in study design, data collection and interpretation, or the decision to submit the work for publication.

### Author contributions

Ellen M Acosta, Conceptualization, Software, Formal analysis, Investigation, Visualization, Methodology, Writing - original draft, Writing – review and editing; Katherine A Little, Investigation,

Methodology; Benjamin P Bratton, Software, Formal analysis, Visualization, Writing – review and editing; Jaime G Lopez, Software, Formal analysis, Writing – review and editing; Xuming Mao, Investigation; Aimee S Payne, Resources, Methodology, Writing – review and editing; Mohamed Donia, Conceptualization, Methodology, Writing – review and editing; Danelle Devenport, Conceptualization, Resources, Methodology, Writing – review and editing; Zemer Gitai, Conceptualization, Resources, Supervision, Funding acquisition, Methodology, Writing - original draft, Writing – review and editing

## Author ORCIDs
Ellen M Acosta http://orcid.org/0000-0002-6744-8411
Katherine A Little http://orcid.org/0000-0001-6993-5312
Benjamin P Bratton http://orcid.org/0000-0003-1128-2560
Aimee S Payne http://orcid.org/0000-0001-9389-7918
Danelle Devenport http://orcid.org/0000-0002-5464-259X
Zemer Gitai http://orcid.org/0000-0002-3280-6178

## Ethics

Human subjects: Human subject research was conducted under Princeton University IRB #13003 at the Princeton University Department of Molecular Biology. Healthy volunteers were recruited via informational flyers. Volunteers gave informed consent prior to sample collection.

All mice were housed and maintained in a certified animal facility and all experiments were conducted according to USA Public Health Service Policy of Humane Care and Use of Laboratory Animals. All protocols were approved by the Institutional Animal Care and Use Committee, protocol #1867 (Princeton University).

Reviewer #1 (Public Review): https://doi.org/10.7554/eLife.87192.2.sa1
Reviewer #2 (Public Review): https://doi.org/10.7554/eLife.87192.2.sa2

## Additional files

### Supplementary files

- Supplementary file 1. Nucleotide sequences used in this study.
- Supplementary file 2. Sequence counts for *Figures 2–4*.
- Supplementary file 3. Sequence counts in perturbation recovery.
- MDAR checklist

### Data availability

Sequencing datasets generated and analyzed during this study are available at NCBI Sequence Read Archive (SRA) (BioProject numbers PRINA918671 and PRINA918959). Oligonucleotide sequences used are included in Table S1.

The following datasets were generated:

| Author(s) | Year | Dataset title | Dataset URL | Database and Identifier |
|---|---|---|---|---|
| Acosta E | 2023 | Human skin microbiome perturbation recovery | https://www.ncbi.nlm.nih.gov/bioproject/PRJNA918671 | NCBI BioProject, PRJNA918671 |
| Acosta E | 2023 | Human skin microbiome | https://www.ncbi.nlm.nih.gov/bioproject/PRJNA918959 | NCBI BioProject, PRJNA918959 |

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
