## [Editor Report · eLife assessment]

In this **important** study, the authors provide **convincing** evidence that current DNA-based microbial genomics for skin bacteria cannot always detect the source of sequenced DNA and whether it originated from viable or non-viable bacteria. Additionally, the authors demonstrated in humans and mice that most of the viable bacteria reside inside hair follicles rather than the surface of the skin per se. Overall, the work has significance beyond a single discipline and will be of interest to those studying microbiomes.

---

## [Referee Report · Reviewer #1 (Public Review)]

The authors of this well-described publication provided strong evidence that current DNA-based microbial genomics methodologies have an inherent constraint. These approaches cannot detect the source of sequenced DNA, and they fail to demonstrate the origin of sequenced DNA from live or non-viable bacteria. Moreover, scientists proved in people and mice that live bacteria for the most part remained within hair follicles rather than on the skin's surface. Overall, this study is of excellent quality and has broad implications beyond a particular subject.

Strengths:

The study is well-designed, and the experimental methods are well-described.

The results are presented clearly and are supported by statistical analyses.

The study's findings are novel and have important implications for understanding the skin microbiome and the biology of the skin.

Weakness:

RNA-based NGS could parallelly study the results of this DNA-based microbiome study. The bulk RNA-Seq can sequence thousands of transcripts from each viable bacterium and match them with the bacterial genome and transcriptome references. It is one of the best confirmatory methods for showing the diversity of viable cutaneous bacteria.

---

## [Referee Report · Reviewer #2 (Public Review)]

The study by Acosta et al. is very interesting as it presents a simple and easy method for identifying live and dead bacteria DNA in the skin - PMA labeling, verified by FISH. This study provides several meaningful conclusions that could inform future skin microbiome studies:

Firstly, the 16s rRNA gene sequencing of skin microbial samples collected by cotton swabs may include DNA from a large number of dead bacteria, leading to an over-representation of skin bacteria in the analysis.

Secondly, the study found that there were fewer live bacteria on the skin surface than the detected bacterial DNA predicted, with most skin bacteria harboring in the hair follicles. This conclusion aligns with the physiological properties of the skin, as the hair follicle epithelium creates a moist, nutrient-rich, low-UV, and immune-privileged environment, which is conducive to the growth, colonization, and development of microorganisms.

Finally, the authors propose that the bacteria on the skin surface originate from the proliferation and replenishment of hair follicle resident bacteria, which could be one reason for the short-term instability and long-term stability of the skin microbiome.

Overall, this study provides valuable insights into the composition and distribution of skin bacteria and highlights the importance of using appropriate methods to identify live bacteria in skin microbiome studies.